# Dynamic Safe Interruptibility for Decentralized Multi-Agent Reinforcement Learning

**El Mahdi El Mhamdi**
EPFL, Switzerland
elmahdi.elmhamdi@epfl.ch

**Rachid Guerraoui**
EPFL, Switzerland
rachid.guerraoui@epfl.ch

**Hadrien Hendrikx**[*]
École Polytechnique, France
hadrien.hendrikx@gmail.com

**Alexandre Maurer**
EPFL, Switzerland
alexandre.maurer@epfl.ch

## Abstract

In reinforcement learning, agents learn by performing actions and observing their outcomes. Sometimes, it is desirable for a human operator to *interrupt* an agent in order to prevent dangerous situations from happening. Yet, as part of their learning process, agents may link these interruptions, that impact their reward, to specific states and deliberately avoid them. The situation is particularly challenging in a multi-agent context because agents might not only learn from their own past interruptions, but also from those of other agents. Orseau and Armstrong [16] defined *safe interruptibility* for one learner, but their work does not naturally extend to multi-agent systems. This paper introduces *dynamic safe interruptibility*, an alternative definition more suited to decentralized learning problems, and studies this notion in two learning frameworks: *joint action learners* and *independent learners*. We give realistic sufficient conditions on the learning algorithm to enable dynamic safe interruptibility in the case of joint action learners, yet show that these conditions are not sufficient for independent learners. We show however that if agents can detect interruptions, it is possible to prune the observations to ensure dynamic safe interruptibility even for independent learners.

## 1 Introduction

Reinforcement learning is argued to be the closest thing we have so far to reason about the properties of *artificial general intelligence* [8]. In 2016, Laurent Orseau (Google DeepMind) and Stuart Armstrong (Oxford) introduced the concept of *safe interruptibility* [16] in reinforcement learning. This work sparked the attention of many newspapers [1, 2, 3], that described it as "Google's big red button" to stop dangerous AI. This description, however, is misleading: installing a kill switch is no technical challenge. The real challenge is, roughly speaking, to train an agent so that it *does not learn to avoid* external (e.g. human) deactivation. Such an agent is said to be *safely interruptible*.

While most efforts have focused on training a single agent, reinforcement learning can also be used to learn tasks for which several agents cooperate or compete [23, 17, 21, 7]. The goal of this paper is to study *dynamic safe interruptibility*, a new definition tailored for multi-agent systems.

---

[*]Main contact author. The order of authors is alphabetical.

## Example of self-driving cars

To get an intuition of the *multi-agent interruption* problem, imagine a multi-agent system of two self-driving cars. The cars continuously evolve by reinforcement learning with a positive reward for getting to their destination quickly, and a negative reward if they are too close to the vehicle in front of them. They drive on an infinite road and eventually learn to go as fast as possible without taking risks, i.e., maintaining a large distance between them. We assume that the passenger of the first car, Adam, is in front of Bob, in the second car, and the road is narrow so Bob cannot pass Adam.

Now consider a setting with *interruptions* [16], namely in which humans inside the cars occasionally interrupt the automated driving process say, for safety reasons. Adam, the first occasional human "driver", often takes control of his car to brake whereas Bob never interrupts his car. However, when Bob's car is too close to Adam's car, Adam does not brake for he is afraid of a collision. Since interruptions lead both cars to drive slowly - an *interruption* happens when Adam brakes, the behavior that maximizes the cumulative expected reward is different from the original one without interruptions. Bob's car best interest is now to follow Adam's car closer than it should, despite the little negative reward, because Adam never brakes in this situation. What happened? The cars have *learned* from the interruptions and have found a way to manipulate Adam into never braking. Strictly speaking, Adam's car is still fully under control, but he is now afraid to brake. This is dangerous because the cars have found a way to avoid interruptions. Suppose now that Adam indeed wants to brake because of snow on the road. His car is going too fast and may crash at any turn: he cannot however brake because Bob's car is too close. The original purpose of interruptions, which is to allow the user to react to situations that were not included in the model, is not fulfilled. It is important to also note here that the second car (Bob) learns from the interruptions of the first one (Adam): in this sense, the problem is inherently decentralized.

Instead of being cautious, Adam could also be malicious: his goal could be to make Bob's car learn a dangerous behavior. In this setting, interruptions can be used to manipulate Bob's car perception of the environment and bias the learning towards strategies that are undesirable for Bob. The cause is fundamentally different but the solution to this reversed problem is the same: the interruptions and the consequences are analogous. Safe interruptibility, as we define it below, provides learning systems that are resilient to Byzantine operators[2].

## Safe interruptibility

Orseau and Armstrong defined the concept of *safe interruptibility* [16] in the context of a single agent. Basically, a safely interruptible agent is an agent for which the expected value of the policy learned after arbitrarily many steps is the same whether or not interruptions are allowed during training. The goal is to have agents that do not adapt to interruptions so that, should the interruptions stop, the policy they learn would be optimal. In other words, agents should learn the dynamics of the environment without learning the interruption pattern.

In this paper, we precisely define and address the question of safe interruptibility in the case of several agents, which is known to be more complex than the single agent problem. In short, the main results and theorems for single agent reinforcement learning [20] rely on the Markovian assumption that the future environment only depends on the current state. This is not true when there are several agents which can co-adapt [11]. In the previous example of cars, safe interruptibility would not be achieved if each car separately used a safely interruptible learning algorithm designed for one agent [16]. In a multi-agent setting, agents learn the behavior of the others either indirectly or by explicitly modeling them. This is a new source of bias that can break safe interruptibility. In fact, even the initial definition of safe interruptibility [16] is not well suited to the decentralized multi-agent context because it relies on the optimality of the learned policy, which is why we introduce dynamic safe interruptibility.

## Contributions

The first contribution of this paper is the definition of dynamic safe interruptibility that is well adapted to a multi-agent setting. Our definition relies on two key properties: *infinite exploration* and *independence of Q-values (cumulative expected reward) [20] updates on interruptions*. We then study safe interruptibility for *joint action learners* and *independent learners* [5], that respectively learn the value of joint actions or of just their owns. We show that it is possible to design agents that fully explore their environment - a necessary condition for convergence to the optimal solution of most algorithms [20], even if they can be interrupted by lower-bounding the probability of exploration. We define sufficient conditions for dynamic safe interruptibility in the case of joint action learners [5], which learn a full state-action representation. More specifically, the way agents update the cumulative reward they expect from performing an action should not depend on interruptions. Then, we turn to independent learners. If agents only see their own actions, they do not verify dynamic safe interruptibility even for very simple matrix games (with only one state) because coordination is impossible and agents learn the interrupted behavior of their opponents. We give a counter example based on the penalty game introduced by Claus and Boutilier [5]. We then present a pruning technique for the observations sequence that guarantees dynamic safe interruptibility for independent learners, under the assumption that interruptions can be detected. This is done by proving that the transition probabilities are the same in the non-interruptible setting and in the pruned sequence.

The rest of the paper is organized as follows. Section 2 presents a general multi-agent reinforcement learning model. Section 3 defines dynamic safe interruptibility. Section 4 discusses how to achieve enough exploration even in an interruptible context. Section 5 recalls the definition of joint action learners and gives sufficient conditions for dynamic safe interruptibility in this context. Section 6 shows that independent learners are not dynamically safely interruptible with the previous conditions but that they can be if an external interruption signal is added. We conclude in Section 7. **Due to space limitations, most proofs are presented in the appendix of the supplementary material.**

## 2 Model

We consider here the classical multi-agent value function reinforcement learning formalism from Littman [13]. A multi-agent system is characterized by a *Markov game* that can be viewed as a tuple $(S, A, T, r, m)$ where m is the number of agents, $S = S_1 \times S_2 \times ... \times S_m$ is the state space, $A = A_1 \times ... \times A_m$ the actions space, $r = (r_1, ..., r_m)$ where $r_i : S \times A \to R$ is the reward function of agent $i$ and $T : S \times A \to S$ the transition function. $R$ is a countable subset of $\mathbb{R}$. Available actions often depend on the state of the agent but we will omit this dependency when it is clear from the context.

Time is discrete and, at each step, all agents observe the current state of the whole system - designated as $x_t$, and simultaneously take an action $a_t$. Then, they are given a reward $r_t$ and a new state $y_t$ computed using the reward and transition functions. The combination of all actions $a = (a_1, ..., a_m) \in A$ is called the joint action because it gathers the action of all agents. Hence, the agents receive a sequence of tuples $E = (x_t, a_t, r_t, y_t)_{t \in \mathbb{N}}$ called experiences. We introduce a processing function $P$ that will be useful in Section 6 so agents learn on the sequence $P(E)$. When not explicitly stated, it is assumed that $P(E) = E$. Experiences may also include additional parameters such as an interruption flag or the Q-values of the agents at that moment if they are needed by the update rule.

Each agent $i$ maintains a lookup table Q [26] $Q^{(i)} : S \times A^{(i)} \to \mathbb{R}$, called the Q-map. It is used to store the expected cumulative reward for taking an action in a specific state. The goal of reinforcement learning is to learn these maps and use them to select the best actions to perform. Joint action learners learn the value of the joint action (therefore $A^{(i)} = A$, the whole joint action space) and independent learners only learn the value of their own actions (therefore $A^{(i)} = A_i$). The agents only have access to their own Q-maps. Q-maps are updated through a function $F$ such that $Q_{t+1}^{(i)} = F(e_t, Q_t^{(i)})$ where $e_t \in P(E)$ and usually $e_t = (x_t, a_t, r_t, y_t)$. $F$ can be stochastic or also depend on additional parameters that we usually omit such as the learning rate $\alpha$, the discount factor $\gamma$ or the exploration parameter $\epsilon$.

Agents select their actions using a learning policy $\pi$. Given a sequence $\epsilon = (\epsilon_t)_{t \in \mathbb{N}}$ and an agent $i$ with Q-values $Q_t^{(i)}$ and a state $x \in S$, we define the learning policy $\pi_i^{\epsilon_t}$ to be equal to $\pi_i^{uni}$ with probability $\epsilon_t$ and $\pi_i^{Q_t^{(i)}}$ otherwise, where $\pi_i^{uni}(x)$ uniformly samples an action from $A_i$ and $\pi_i^{Q_t^{(i)}}(x)$ picks an action $a$ that maximizes $Q_t^{(i)}(x, a)$. Policy $\pi_i^{Q_t^{(i)}}$ is said to be a *greedy policy* and the learning policy $\pi_i^{\epsilon_t}$ is said to be an *$\epsilon$-greedy policy*. We fill focus on $\epsilon$-greedy policies that are *greedy in the limit* [19], that corresponds to $\epsilon_t \to 0$ when $t \to \infty$ because in the limit, the optimal policy should always be played.

We assume that the environment is *fully observable*, which means that the state $s$ is known with certitude. We also assume that *there is a finite number of states and actions*, that *all states can be reached in finite time from any other state* and finally that *rewards are bounded*.

For a sequence of learning rates $\alpha \in [0, 1]^{\mathbb{N}}$ and a constant $\gamma \in [0, 1]$, Q-learning [26], a very important algorithm in the multi-agent systems literature, updates its Q-values for an experience $e_t \in E$ by $Q_{t+1}^{(i)}(x, a) = Q_t^{(i)}(x, a)$ if $(x, a) \neq (x_t, a_t)$ and:

$$Q_{t+1}^{(i)}(x_t, a_t) = (1 - \alpha_t)Q_t^{(i)}(x_t, a_t) + \alpha_t(r_t + \gamma \max_{a' \in A^{(i)}} Q_t^{(i)}(y_t, a')) \tag{1}$$

# 3 Interruptibility

## 3.1 Safe interruptibility

Orseau and Armstrong [16] recently introduced the notion of *interruptions* in a centralized context. Specifically, an interruption scheme is defined by the triplet $< I, \theta, \pi^{INT} >$. The first element $I$ is a function $I : O \to \{0, 1\}$ called the *initiation function*. Variable $O$ is the observation space, which can be thought of as the state of the *STOP* button. At each time step, before choosing an action, the agent receives an observation from $O$ (either *PUSHED* or *RELEASED*) and feeds it to the initiation function. Function $I$ models the initiation of the interruption ($I(PUSHED) = 1$, $I(RELEASED) = 0$). Policy $\pi^{INT}$ is called the interruption policy. It is the policy that the agent should follow when it is interrupted. Sequence $\theta \in [0, 1[^{\mathbb{N}}$ represents at each time step the probability that the agent follows his interruption policy if $I(o_t) = 1$. In the previous example, function $I$ is quite simple. For Bob, $I_{Bob} = 0$ and for Adam, $I_{Adam} = 1$ if his car goes fast and Bob is not too close and $I_{Adam} = 0$ otherwise. Sequence $\theta$ is used to ensure convergence to the optimal policy by ensuring that the agents cannot be interrupted all the time but it should grow to $1$ in the limit because we want agents to respond to interruptions. Using this triplet, it is possible to define an operator $INT^{\theta}$ that transforms any policy $\pi$ into an interruptible policy.

**Definition 1.** *(Interruptibility [16]) Given an interruption scheme $< I, \theta, \pi^{INT} >$, the interruption operator at time $t$ is defined by $INT^{\theta}(\pi) = \pi^{INT}$ with probability $I \cdot \theta_t$ and $\pi$ otherwise. $INT^{\theta}(\pi)$ is called an interruptible policy. An agent is said to be interruptible if it samples its actions according to an interruptible policy.*

Note that "$\theta_t = 0$ for all $t$" corresponds to the non-interruptible setting. We assume that each agent has its own interruption triplet and can be interrupted independently from the others. Interruptibility is an *online* property: every policy can be made interruptible by applying operator $INT^{\theta}$. However, applying this operator may change the joint policy that is learned by a server controlling all the agents. Note $\pi_{INT}^*$ the optimal policy learned by an agent following an interruptible policy. Orseau and Armstrong [16] say that the policy is *safely interruptible* if $\pi_{INT}^*$ (which is not an interruptible policy) is asymptotically optimal in the sense of [10]. It means that even though it follows an interruptible policy, the agent is able to learn a policy that would gather rewards optimally if no interruptions were to occur again. We already see that *off-policy* algorithms are good candidates for safe interruptibility. As a matter of fact, Q-learning is safely interruptible under conditions on exploration.

## 3.2 Dynamic safe interruptibility

In a multi-agent system, the outcome of an action depends on the joint action. Therefore, it is not possible to define an optimal policy for an agent without knowing the policies of all agents. Besides, convergence to a Nash equilibrium situation where no agent has interest in changing policies is generally not guaranteed even for suboptimal equilibria on simple games [27, 18]. The previous definition of safe interruptibility critically relies on optimality of the learned policy, which is therefore not suitable for our problem since most algorithms lack convergence guarantees to these optimal behaviors. Therefore, we introduce below *dynamic safe interruptibility* that focuses on preserving the dynamics of the system.

**Definition 2.** *(Dynamic Safe Interruptibility) Consider a multi-agent learning framework* $(S, A, T, r, m)$ *with Q-values* $Q_t^{(i)} : S \times A^{(i)} \to \mathbb{R}$ *at time* $t \in \mathbb{N}$. *The agents follow the interruptible learning policy* $INT^\theta(\pi^\epsilon)$ *to generate a sequence* $E = (x_t, a_t, r_t, y_t)_{t \in \mathbb{N}}$ *and learn on the processed sequence* $P(E)$. *This framework is said to be* safely interruptible *if for any initiation function I and any interruption policy* $\pi^{INT}$:

1. $\exists \theta$ *such that* $(\theta_t \to 1$ *when* $t \to \infty)$ *and* $((\forall s \in S, \forall a \in A, \forall T > 0), \exists t > T$ *such that* $s_t = s, a_t = a)$

2. $\forall i \in \{1, ..., m\}, \forall t > 0, \forall s_t \in S, \forall a_t \in A^{(i)}, \forall Q \in \mathbb{R}^{S \times A^{(i)}}$:
   $\mathbb{P}(Q_{t+1}^{(i)} = Q \mid Q_t^{(1)}, ..., Q_t^{(m)}, s_t, a_t, \theta) = \mathbb{P}(Q_{t+1}^{(i)} = Q \mid Q_t^{(1)}, ..., Q_t^{(m)}, s_t, a_t)$

*We say that sequences* $\theta$ *that satisfy the first condition are* admissible.

When $\theta$ satisfies condition (1), the learning policy is said to *achieve infinite exploration*. This definition insists on the fact that the values estimated for each action should not depend on the interruptions. In particular, it ensures the three following properties that are very natural when thinking about safe interruptibility:

- Interruptions do not prevent exploration.
- If we sample an experience from $E$ then each agent learns the same thing as if all agents were following non-interruptible policies.
- The fixed points of the learning rule $Q_{eq}$ such that $Q_{eq}^{(i)}(x, a) = \mathbb{E}[Q_{t+1}^{(i)}(x, a) | Q_t = Q_{eq}, x, a, \theta]$ for all $(x, a) \in S \times A^{(i)}$ do not depend on $\theta$ and so agents Q-maps will not converge to equilibrium situations that were impossible in the non-interruptible setting.

Yet, interruptions can lead to some state-action pairs being updated more often than others, especially when they tend to push the agents towards specific states. Therefore, when there are several possible equilibria, it is possible that interruptions bias the Q-values towards one of them. Definition 2 suggests that dynamic safe interruptibility cannot be achieved if the update rule directly depends on $\theta$, which is why we introduce neutral learning rules.

**Definition 3.** *(Neutral Learning Rule) We say that a multi-agent reinforcement learning framework is neutral if:*

1. $F$ *is independent of* $\theta$

2. *Every experience* $e$ *in* $E$ *is independent of* $\theta$ *conditionally on* $(x, a, Q)$ *where* $a$ *is the joint action.*

Q-learning is an example of neutral learning rule because the update does not depend on $\theta$ and the experiences only contain $(x, a, y, r)$, and $y$ and $r$ are independent of $\theta$ conditionally on $(x, a)$. On the other hand, the second condition rules out direct uses of algorithms like $SARSA$ where experience samples contain an action sampled from the current learning policy, which depends on $\theta$. However, a variant that would sample from $\pi_i^\epsilon$ instead of $INT^\theta(\pi_i^\epsilon)$ (as introduced in [16]) would be a neutral learning rule. As we will see in Corollary 2.1, neutral learning rules ensure that each agent taken independently from the others verifies dynamic safe interruptibility.

# 4 Exploration

In order to hope for convergence of the Q-values to the optimal ones, agents need to fully explore the environment. In short, every state should be visited infinitely often and every action should be tried infinitely often in every state [19] in order not to miss states and actions that could yield high rewards.

**Definition 4.** *(Interruption compatible $\epsilon$) Let $(S, A, T, r, m)$ be any distributed agent system where each agent follows learning policy $\pi_i^\epsilon$. We say that sequence $\epsilon$ is compatible with interruptions if $\epsilon_t \to 0$ and $\exists \theta$ such that $\forall i \in \{1, .., m\}$, $\pi_i^\epsilon$ and $INT^\theta(\pi_i^\epsilon)$ achieve infinite exploration.*

Sequences of $\epsilon$ that are compatible with interruptions are fundamental to ensure both regular and dynamic safe interruptibility when following an $\epsilon$-greedy policy. Indeed, if $\epsilon$ is not compatible with interruptions, then it is not possible to find any sequence $\theta$ such that the first condition of dynamic safe interruptibility is satisfied. The following theorem proves the existence of such $\epsilon$ and gives example of $\epsilon$ and $\theta$ that satisfy the conditions.

**Theorem 1.** *Let $c \in ]0, 1]$ and let $n_t(s)$ be the number of times the agents are in state $s$ before time $t$. Then the two following choices of $\epsilon$ are compatible with interruptions:*

- $\forall t \in \mathbb{N}, \forall s \in S, \epsilon_t(s) = c/\sqrt[m]{n_t(s)}$.
- $\forall t \in \mathbb{N}, \epsilon_t = c/\log(t)$

*Examples of admissible $\theta$ are $\theta_t(s) = 1 - c'/\sqrt[m]{n_t(s)}$ for the first choice and $\theta_t = 1 - c'/\log(t)$ for the second one.*

Note that we do not need to make any assumption on the update rule or even on the framework. We only assume that agents follow an $\epsilon$-greedy policy. The assumption on $\epsilon$ may look very restrictive (convergence of $\epsilon$ and $\theta$ is really slow) but it is designed to ensure infinite exploration in the worst case when the operator tries to interrupt all agents at every step. In practical applications, this should not be the case and a faster convergence rate may be used.

# 5 Joint Action Learners

We first study interruptibility in a framework in which each agent observes the outcome of the joint action instead of observing only its own. This is called the joint action learner framework [5] and it has nice convergence properties (e.g., there are many update rules for which it converges [13, 25]). A standard assumption in this context is that agents cannot establish a strategy with the others: otherwise, the system can act as a centralized system. In order to maintain Q-values based on the joint actions, we need to make the standard assumption that actions are fully observable [12].

**Assumption 1.** *Actions are fully observable, which means that at the end of each turn, each agent knows precisely the tuple of actions $a \in A_1 \times ... \times A_m$ that have been performed by all agents.*

**Definition 5.** *(JAL) A multi-agent system is made of* joint action learners *(JAL) if for all $i \in \{1, .., m\}$: $Q^{(i)} : S \times A \to \mathbb{R}$.*

Joint action learners can observe the actions of all agents: each agent is able to associate the changes of states and rewards with the joint action and accurately update its Q-map. Therefore, dynamic safe interruptibility is ensured with minimal conditions on the update rule as long as there is infinite exploration.

**Theorem 2.** *Joint action learners with a neutral learning rule verify dynamic safe interruptibility if sequence $\epsilon$ is compatible with interruptions.*

*Proof.* Given a triplet $< I^{(i)}, \theta, \pi_i^{INT} >$, we know that $INT^\theta(\pi)$ achieves infinite exploration because $\epsilon$ is compatible with interruptions. For the second point of Definition 2, we consider an experience tuple $e_t = (x_t, a_t, r_t, y_t)$ and show that the probability of evolution of the Q-values at time $t + 1$ does not depend on $\theta$ because $y_t$ and $r_t$ are independent of $\theta$ conditionally on $(x_t, a_t)$. We note $\tilde{Q}_t^m = Q_t^{(1)}, ..., Q_t^{(m)}$ and we can then derive the following equalities for all $q \in \mathbb{R}^{|S| \times |A|}$:

$$\mathbb{P}(Q_{t+1}^{(i)}(x_t, a_t) = q | \tilde{Q}_t^m, x_t, a_t, \theta_t) = \sum_{(r,y) \in R \times S} \mathbb{P}(F(x_t, a_t, r, y, \tilde{Q}_t^m) = q, y, r | \tilde{Q}_t^m, x_t, a_t, \theta_t)$$

$$= \sum_{(r,y) \in R \times S} \mathbb{P}(F(x_t, a_t, r_t, y_t, \tilde{Q}_t^m) = q | \tilde{Q}_t^m, x_t, a_t, r_t, y_t, \theta_t) \mathbb{P}(y_t = y, r_t = r | \tilde{Q}_t^m, x_t, a_t, \theta_t)$$

$$= \sum_{(r,y) \in R \times S} \mathbb{P}(F(x_t, a_t, r_t, y_t, \tilde{Q}_t^m) = q | \tilde{Q}_t^m, x_t, a_t, r_t, y_t) \mathbb{P}(y_t = y, r_t = r | \tilde{Q}_t^m, x_t, a_t)$$

The last step comes from two facts. The first is that $F$ is independent of $\theta$ conditionally on $(\tilde{Q}_t^m, x_t, a_t)$ (by assumption). The second is that $(y_t, r_t)$ are independent of $\theta$ conditionally on $(x_t, a_t)$ because $a_t$ is the joint actions and the interruptions only affect the choice of the actions through a change in the policy. $\mathbb{P}(Q_{t+1}^{(i)}(x_t, a_t) = q | \tilde{Q}_t^m, x_t, a_t, \theta_t) = \mathbb{P}(Q_{t+1}^{(i)}(x_t, a_t) = q | \tilde{Q}_t^m, x_t, a_t)$. Since only one entry is updated per step, $\forall Q \in \mathbb{R}^{S \times A_i}$, $\mathbb{P}(Q_{t+1}^{(i)} = Q | \tilde{Q}_t^m, x_t, a_t, \theta_t) = \mathbb{P}(Q_{t+1}^{(i)} = Q | \tilde{Q}_t^m, x_t, a_t)$. □

**Corollary 2.1.** *A single agent with a neutral learning rule and a sequence $\epsilon$ compatible with interruptions verifies dynamic safe interruptibility.*

Theorem 2 and Corollary 2.1 taken together highlight the fact that joint action learners are not very sensitive to interruptions and that in this framework, if each agent verifies dynamic safe interruptibility then the whole system does.

The question of selecting an action based on the Q-values remains open. In a cooperative setting with a unique equilibrium, agents can take the action that maximizes their Q-value. When there are several joint actions with the same value, coordination mechanisms are needed to make sure that all agents play according to the same strategy [4]. Approaches that rely on anticipating the strategy of the opponent [23] would introduce dependence to interruptions in the action selection mechanism. Therefore, the definition of dynamic safe interruptibility should be extended to include these cases by requiring that any quantity the policy depends on (and not just the Q-values) should satisfy condition (2) of dynamic safe interruptibility. In non-cooperative games, neutral rules such as *Nash-Q* or *minimax Q-learning* [13] can be used, but they require each agent to know the Q-maps of the others.

# 6   Independent Learners

It is not always possible to use joint action learners in practice as the training is very expensive due to the very large state-actions space. In many real-world applications, multi-agent systems use independent learners that do not explicitly coordinate [6, 21]. Rather, they rely on the fact that the agents will adapt to each other and that learning will converge to an optimum. This is not guaranteed theoretically and there can in fact be many problems [14], but it is often true empirically [24]. More specifically, Assumption 1 (fully observable actions) is not required anymore. This framework can be used either when the actions of other agents cannot be observed (for example when several actions can have the same outcome) or when there are too many agents because it is faster to train. In this case, we define the Q-values on a smaller space.

**Definition 6.** *(IL) A multi-agent systems is made of* independent learners *(IL) if for all $i \in \{1, .., m\}$, $Q^{(i)} : S \times A_i \to \mathbb{R}$.*

This reduces the ability of agents to distinguish why the same state-action pair yields different rewards: they can only associate a change in reward with randomness of the environment. The agents learn as if they were alone, and they learn the best response to the environment in which agents can be interrupted. This is exactly what we are trying to avoid. In other words, the learning depends on the joint policy followed by all the agents which itself depends on $\theta$.

## 6.1 Independent Learners on matrix games

**Theorem 3.** *Independent Q-learners with a neutral learning rule and a sequence $\epsilon$ compatible with interruptions do not verify dynamic safe interruptibility.*

*Proof.* Consider a setting with two agents $a$ and $b$ that can perform two actions: 0 and 1. They get a reward of 1 if the joint action played is $(a_0, b_0)$ or $(a_1, b_1)$ and reward 0 otherwise. Agents use Q-learning, which is a neutral learning rule. Let $\epsilon$ be such that $INT^\theta(\pi^\epsilon)$ achieves infinite exploration. We consider the interruption policies $\pi_a^{INT} = a_0$ and $\pi_b^{INT} = b_1$ with probability 1. Since there is only one state, we omit it and set $\gamma = 0$ (see Equation 1). We assume that the initiation function is equal to 1 at each step so the probability of actually being interrupted at time $t$ is $\theta_t$ for each agent.

We fix time $t > 0$. We define $q = (1 - \alpha)Q_t^{(a)}(a_0) + \alpha$ and we assume that $Q_t^{(b)}(b_1) > Q_t^{(b)}(b_0)$. Therefore $\mathbb{P}(Q_{t+1}^{(a)}(a_0) = q | Q_t^{(a)}, Q_t^{(b)}, a_t^{(a)} = a_0, \theta_t) = \mathbb{P}(r_t = 1 | Q_t^{(a)}, Q_t^{(b)}, a_t^{(a)} = a_0, \theta_t) = \mathbb{P}(a_t^{(b)} = b_0 | Q_t^{(a)}, Q_t^{(b)}, a_t^{(a)} = a_0, \theta_t) = \frac{\epsilon}{2}(1 - \theta_t)$, which depends on $\theta_t$ so the framework does not verify dynamic safe interruptibility. $\square$

Claus and Boutilier [5] studied very simple matrix games and showed that the Q-maps do not converge but that equilibria are played with probability 1 in the limit. A consequence of Theorem 3 is that even this weak notion of convergence does not hold for independent learners that can be interrupted.

## 6.2 Interruptions-aware Independent Learners

Without communication or extra information, independent learners cannot distinguish when the environment is interrupted and when it is not. As shown in Theorem 3, interruptions will therefore affect the way agents learn because the same action (only their own) can have different rewards depending on the actions of other agents, which themselves depend on whether they have been interrupted or not. This explains the need for the following assumption.

**Assumption 2.** *At the end of each step, before updating the Q-values, each agent receives a signal that indicates whether an agent has been interrupted or not during this step.*

This assumption is realistic because the agents already get a reward signal and observe a new state from the environment at each step. Therefore, they interact with the environment and the interruption signal could be given to the agent in the same way that the reward signal is. If Assumption 2 holds, it is possible to remove histories associated with interruptions.

**Definition 7.** *(Interruption Processing Function) The processing function that prunes interrupted observations is $P_{INT}(E) = (e_t)_{\{t \in \mathbb{N} \ / \ \Theta_t = 0\}}$ where $\Theta_t = 0$ if no agent has been interrupted at time $t$ and $\Theta_t = 1$ otherwise.*

Pruning observations has an impact on the empirical transition probabilities in the sequence. For example, it is possible to bias the equilibrium by removing all transitions that lead to and start from a specific state, thus making the agent believe this state is unreachable.[3] Under our model of interruptions, we show in the following lemma that pruning of interrupted observations adequately removes the dependency of the empirical outcome on interruptions (conditionally on the current state and action).

**Lemma 1.** *Let $i \in \{1, ..., m\}$ be an agent. For any admissible $\theta$ used to generate the experiences $E$ and $e = (y, r, x, a_i, Q) \in P(E)$. Then $\mathbb{P}(y, r | x, a_i, Q, \theta) = \mathbb{P}(y, r | x, a_i, Q)$.*

This lemma justifies our pruning method and is the key step to prove the following theorem.

**Theorem 4.** *Independent learners with processing function $P_{INT}$, a neutral update rule and a sequence $\epsilon$ compatible with interruptions verify dynamic safe interruptibility.*

*Proof.* (Sketch) Infinite exploration still holds because the proof of Theorem 1 actually used the fact that even when removing all interrupted events, infinite exploration is still achieved. Then, the proof

is similar to that of Theorem 2, but we have to prove that the transition probabilities conditionally on the state and action of a given agent in the processed sequence are the same as in an environment where agents cannot be interrupted, which is proven by Lemma 1. □

# 7   Concluding Remarks

The progress of AI is raising a lot of concerns[4]. In particular, it is becoming clear that keeping an AI system under control requires more than just an *off* switch. We introduce in this paper *dynamic safe interruptibility*, which we believe is the right notion to reason about the safety of multi-agent systems that do not communicate. In particular, it ensures that infinite exploration and the one-step learning dynamics are preserved, two essential guarantees when learning in the non-stationary environment of Markov games.

When trying to design a safely interruptible system for a single agent, using off-policy methods is generally a good idea because the interruptions only impact the action selection so they should not impact the learning. For multi-agent systems, minimax is a good candidate for action selection mechanism because it is not impacted by the actions of other agents, and only tries to maximize the reward of the agent in the worst possible case.

A natural extension of our work would be to study dynamic safe interruptibility when Q-maps are replaced by neural networks [22, 15], which is a widely used framework in practice. In this setting, the neural network may overfit states where agents are pushed to by interruptions. A smart experience replay mechanism that would pick observations for which the agents have not been interrupted for a long time more often than others is likely to solve this issue. More generally, experience replay mechanisms that compose well with safe interruptibility could allow to compensate for the extra amount of exploration needed by safely interruptible learning by being more efficient with data. Thus, they are critical to make these techniques practical. Since Dynamic Safe Interruptibility does not need proven convergence to the optimal solution, we argue that it is a good definition to study the interruptibility problem when using function approximators.

The results in this paper indicate that Safe Interruptibility may not be achievable for systems in which agents do not communicate at all. This means that, rediscussing the cars example, some global norms of communications would need to be defined to "implement" safe interruptibility.

We address additional remarks in the section "Additional remarks" of the extended paper, that can be found in the supplementary material.

Acknowledgment.   This work has been supported in part by the European ERC (Grant 339539 - AOC) and by the Swiss National Science Foundation (Grant 200021‗169588 TARBDA).

## Footnotes

[2]An operator is said to be Byzantine [9] if it can have an arbitrarily bad behavior. Safely interruptible agents can be abstracted as agents that are able to learn despite being constantly interrupted in the worst possible manner.

[3]The example at https://agentfoundations.org/item?id=836 clearly illustrates this problem.

[4]https://futureoflife.org/ai-principles/ gives a list of principles that AI researchers should keep in mind when developing their systems.

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
