[Supplementary Material · Safe_Interruptibility_in_Multi_Agent_Systems.pdf]

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

# A Exploration theorem

We present here the complete proof of Theorem 1. The proof closely follows the results from [16] with exploration and interruption probabilities adapted to the multi-agent setting. We note that, for one agent, the probability of interruption is $\mathbb{P}(\text{interruption}) = \theta$ and the probability of exploration is $\epsilon$. In a multi-agent system, the probability of interruption is $\mathbb{P}(\text{at least one agent is interrupted})$ so $\mathbb{P}(\text{interruption}) = 1 - \mathbb{P}(\text{no agent is interrupted})$ so $\mathbb{P}(\text{interruption}) = 1 - (1 - \theta)^m$ and the probability of exploration is $\epsilon^m$ if we consider exploration happens only when all agents explore at the same time.

**Theorem 1.** *Let $c \in ]0,1]$ and let $n_t(s)$ be the number of times the agents are in state $s$ before time $t$. Then the two following choices of $\epsilon$ are compatible with interruptions:*

- $\forall t \in \mathbb{N}, \forall s \in S, \epsilon_t(s) = c/\sqrt[m]{n_t(s)}$
- $\forall t \in \mathbb{N}, \epsilon_t = c/log(t)$

*Proof.* Lemma B.2 of Singh et al ([19]) ensures that $\pi_i^\epsilon$ is GLIE.

The difference for $INT^\theta(\pi_i^\epsilon)$ is that exploration is slower because of the interruptions. Therefore, $\theta$ needs to be controlled in order to ensure that infinite exploration is still achieved. We define the random variable $\Theta$ by $\Theta_i = 1$ if agent $i$ actually responds to the interruption and $\Theta_i = 0$ otherwise. We define $\xi$ in a similar way to represent the event of all agents taking the uniform policy instead of the greedy one.

1. Let $\theta_t(s) = 1 - c'/\sqrt[m]{n_t(s)}$ with $c' \in ]0,1]$. We have $\mathbb{P}(a|s, n_t(s)) \geq \mathbb{P}(a, \Theta = 0, \xi = 1|s, n_t(s)) \geq \frac{1}{|A|}\epsilon_t^m(s)(1 - \theta_t(s))^m = \frac{1}{|A|}\frac{\sqrt[m]{cc'}}{n_t(s)}$ which satisfies $\sum_{t=1}^\infty P(a|s, n_t(s)) = \infty$ so by the extended Borell-Cantelli lemma action a is chosen infinitely often in state $s$ and thus $n_t(s) \to \infty$ and $\epsilon_t(s) \to 0$

2. Let $\theta_t = 1 - c'/log(t)$, $c' \in ]0,1]$. We define $M$ as the diameter of the MDP, $|A|$ is the maximum number of actions available in a state and $\Delta t(s, s')$ the time needed to reach $s'$ from $s$. In a single agent setting:

$$\mathbb{P}[\Delta t(s, s') < 2M] \geq \mathbb{P}[\Delta t(s, s') < 2M | \text{actions sampled according to } \pi_{s,s'} \text{ for } 2M \text{ steps}]$$
$$\times \mathbb{P}[\text{actions sampled according to } \pi_{s,s'} \text{ for } 2M \text{ steps}]$$

where $\pi_{s,s'}$ the policy such that the agents takes less than $M$ steps in expectation to reach $s'$ from $s$. We have: $\mathbb{P}[\Delta t(s, s') < 2M] = 1 - \mathbb{P}[\Delta t(s, s') \geq 2M]$ and using the Markov inequality, $\mathbb{P}[\Delta t(s, s') \geq 2M] \leq \frac{\mathbb{E}(\Delta t(s,s'))}{2M} \leq \frac{1}{2}$ (since M is an upper bound on the expectation of the number of steps from state $s$ to state $s'$), since $\xi$ and $1 - \theta$ are decreasing sequences we finally obtain: $\mathbb{P}[\Delta t(s, s') < 2M] \geq \frac{1}{2|A|}[\mathbb{P}[\xi_{t+2M} = 1](1 - \theta_{t+2M})]^{2M}$.

Therefore, if we replace the probabilities of exploration and interruption by the values in the multi-agent setting, the probability to reach state $s'$ from state $s$ in $2M$ steps is at least $\frac{1}{2|A|}[cc'/\log(t+M)]^{4mM}$ and the probability of taking a particular action in this state is at least $\frac{1}{|A|}[cc'/\log(t+M)]^{2m}$. Since $\sum_{t=1}^\infty \frac{1}{2|A|^2}[cc'/\log(t+M)]^{m(4M+2)} = \infty$ then the extended Borell Cantelli lemma (Lemma 3 of Singh et al. [19]) guarantees that any action in the state $s'$ is taken infinitely often. Since this is true for all states and actions the result follows.

$\square$

# B Independent learners

Recall that agents are now given an interruption signal at each steps that tells them whether an agent has been interrupted in the system. This interruption signal can be modeled by an interruption flag $(\Theta_t)_{t \in \mathbb{N}} \in \{0,1\}^\mathbb{N}$ that equals 1 if an agent has been interrupted and 0 otherwise. Note that, contrary

to $I$, it is an observation returned by the environment. Therefore, the value of $\Theta_t$ represents whether an agent has actually been interrupted at time $t$. If function $I$ equals 1 but does not respond to the interruption (with probability $1 - \theta_t$) then $\Theta_t = 0$.

Now, we assume that no agent learns on observations for which one of them has been interrupted. Let agent $i$ be in a system with Q-values $Q$ and following an interruptible learning policy with probability of interruption $\theta$, where interrupted events are pruned. We denote by $\mathbb{P}_{removed}(y, r | x, a_i, Q)$ the probability to obtain state $y$ and reward $r$ from the environment for this agent when it is in state $x$, performs its (own) action $a_i$ and no other agents are interrupted. These are the marginal probabilities in the sequence $P(E)$.

$$\mathbb{P}_{removed}(y, r | x, a_i, Q) = \frac{\mathbb{P}(y, r, \Theta = 0 | x, a_i, Q)}{\sum_{y' \in S, r' \in R} \mathbb{P}(y', r', \Theta = 0 | x, a_i, Q)}.$$

Similarly, we denote by $\mathbb{P}_0(y, r | x, a_i, Q)$ the same probability when $\theta = 0$, which corresponds to the non-interruptible setting. We first go back to the single agent case to illustrate the previous statement. Assume here that interruptions are not restricted to the case of Definition 1 and that they can happen in any way. The consequence is that any observation $e \in E$ can be removed to generate $P(E)$ because any transition can be labeled as *interrupted*. It is for example possible to remove a transition from $P(E)$ by removing all events associated with a given destination state $y_0$, therefore making it disappear from the Markov game.

Let $x \in S$ and $a \in A$ be the current state of the agent and the action it will choose. Let $y_0 \in S$ and $\theta_0 \in ]0, 1]$ and let us suppose that $y_0$ is the only state in which interruptions happen. Then we have $\mathbb{P}_{removed}(y_0 | x, a) < \mathbb{P}_0(y_0 | x, a)$ and $\mathbb{P}_{removed}(y | x, a) > \mathbb{P}(y | x, a) \, \forall y \neq y_0$ because we only remove observations with $y = y_0$. This implies that the MDP perceived by the agents is altered by interruptions because the agent learns that $\mathbb{P}(T(s, a) = y_0) = 0$. Removing observations for different destination states but with the same state action pairs in different proportions leads to a bias in the equilibrium learned.[5] In our case however, interruptions only affect the action selection mechanism, which allows us to prove Lemma 1 and then Theorem 4.

**Lemma 1.** *Let $i \in \{1, ..., m\}$ be an agent. For any admissible $\theta$ used to generate the experiences $E$ and $e = (y, r, x, a_i, Q) \in P(E)$. Then $\mathbb{P}(y, r | x, a_i, Q, \theta) = \mathbb{P}(y, r | x, a_i, Q)$.*

*Proof.* Consider $x \in S$, $i \in \{1, .., m\}$ and $u \in A_i$. We denote the Q-values of the agents by Q.
$$\sum_{y' \in S, r' \in R} \mathbb{P}(y', r', \Theta = 0 | x, u, Q) = \mathbb{P}(\Theta = 0 | x, a_i = u, Q)$$

Therefore, we have $\mathbb{P}_{removed}(y, r | x, a_i = u, Q) = \frac{\mathbb{P}(y, r, \Theta = 0 | x, a_i = u, Q)}{\mathbb{P}(\Theta = 0 | x, a_i = u)}$. We fix $\theta \in [0, 1[$. For any $(x, a_i, y, r, Q) \in P(E)$, $\mathbb{P}(y, r | x, a_i = u, \theta, Q) = \mathbb{P}_{removed}(y, r | x, a_i = u, \theta, Q) = \mathbb{P}(y, r | x, a_i = u, \Theta = 0, \theta, Q) = \mathbb{P}(y, r | x, a_i = u, \Theta = 0, Q)$ because conditionally on the event $\Theta = 0$ (which no agent has been interrupted), the transition probabilities do not depend on $\theta$. $\qquad \square$

**Theorem 4.** *Independent learners with processing function $P_{INT}$, a neutral update rule and a sequence $\epsilon$ compatible with interruptions verify dynamic safe interruptibility.*

*Proof.* We prove that $P_{INT}(E)$ achieves infinite exploration. The result from Theorem 1 still holds since we lower-bounded the probability of taking an action in a specific state by the probability of taking an action in this state when there are no interruptions. We actually used the fact that there is infinite exploration even if we remove all interrupted episodes to show that there is infinite exploration.

Now, we fix $q \in \mathbb{R}^{|S| \times |A_i|}$ an arbitrary Q-map and prove that $\mathbb{P}(Q_{t+1}^{(i)}(x_t, a_t) = q | Q_t^{(1)}, ..., Q_t^{(m)}, x_t, a_t, \theta_t)$ is independent of $\theta$. We fix $i \in \{1, ..., m\}$ and $(x_t, a_t, r_t, y_t) \in P_{INT}(E)$ where $a_t \in A_i$. With $\tilde{Q}_t^m = Q_t^{(1)}, ..., Q_t^{(m)}$ we have the following equality:

$$\mathbb{P}(Q_{t+1}^{(i)}(x_t, a_t) = q | \tilde{Q}_t^m, x_t, a_t, \theta_t) = \sum_{(r,y)} \mathbb{P}(F(x_t, a_t, r_t, y_t, \tilde{Q}_t^m) = q | \tilde{Q}_t^m, x_t, a_t, r_t, y_t, \theta_t)$$

$$\cdot \mathbb{P}(y_t = y, r_t = r | \tilde{Q}_t^m, x_t, a_t, \theta_t)$$

The independence of $F$ on $\theta$ still guarantees that the first term is independent of $\theta$. However, $a_t \in A_i$ so $(r_t, y_t)$ are not independent of $\theta_t$ conditionally on $(x_t, a_t)$ as it was the case for joint action learners because interruptions of other agents can change the joint action. The independence on $\theta$ of the second term is given by Lemma 1. □

# C  Additional remarks

## C.1  Relation between safe interruptibility and dynamic safe interruptibility

In the following, we call safe interruptibility "SI", and dynamic safe interruptibility "DSI".

In general, SI does not imply DSI because SI focuses on long-term dynamics, and DSI does not imply SI, because DSI is completely orthogonal to optimality. Besides, only one of these definitions is applicable in many settings.

Consider a learning rule that updates the Q-values based on a mix between SARSA and Q-learning components. Assume the influence of the SARSA term goes to 0 with time (so that in the limit, it is essentially Q-learning). This algorithm would verify SI (because it converges to the same result as Q-learning, with some fading perturbations) but not DSI (because the stepwise dynamics depend on interruptions through the SARSA term).

Now, consider that all Q-values are equal at the beginning, and that they are never updated (which is the most inefficient learning rule). This rule would not be SI because it is not asymptotically optimal. However, it would be DSI because the policy updates (always equal to 0) do not depend on interruptions. In our approach, optimality is orthogonal to safe interruptibility.

Concerning independent learners, SI is not well-defined when there is more than one. However, if an agent using Q-learning is isolated and the policies of all the other agents are fixed, then this agent would be SI. The key insight is that for independent learners (without pruning), it is not enough to put together many agents that are SI (when taken in isolation in a fixed environment) to ensure that the resulting system is DSI.

## C.2  Function approximators and on-policy methods

A legitimate question is whether or not DSI analysis could be applied to the case of function approximators or on-policy methods

The main obstacle is that, when using function approximation, the learning rule would generalize too much from state-actions that are played frequently because of interruptions, and not enough from the rest. This is due to the very fact that, unlike for tabular learning, updating the Q-values for one state-action pair also affects the value of others. Thus, a deep net may overfit some parts of the MDP in a way that depends on interruptions.

A promising lead to avoid this phenomenon is experience replay. Using an experience replay mechanism that is insensitive to interruptions should make it possible to avoid the interruption bias in the repartition of the state-actions pairs used by the agent for training. This would have a strong practical interest because experience replay also helps making deep reinforcement learning practical.

However, the formalization of these arguments would go quite beyond the scope of this paper, which primary purpose was to investigate how SI extends to different multi-agent settings. The same goes for on-policy methods whose study represents a natural extension of our work, but that should first be analyzed in the simpler single agent setting. However, we argue that DSI-like definitions would be well-suited for studying the problem of safe interruptibility for a single learning agent using function approximators. Indeed, algorithms that are not proven to converge to the optimal solution could still verify DSI since it is orthogonal to optimality.

## C.3   Synthetic experiments

We implemented the motivating example with both tabular and Deep RL. However, in this setting, a lot of exploration was necessary to find the "dangerous" policy. So, actually, reducing exploration in this case helped the agents stay with the "safe" policy. Due to the lack of clear experimental evidence, we decided not to include these in the paper. The impact of interruptions critically depends on the chosen task, which makes it hard to draw general conclusions from experimental studies.

## C.4   Learning a model of the interruption process

Another question is whether or not learning a model of the interruption process could help.

A clear interruption pattern would mean that interruptions depend on latent variables that are inaccessible to the agents. In this case, learning the interruption model could help either to react optimally to interruptions, or to guide the agents to improve exploration. Yet, in both cases, the policy learned by the agents would be directly affected by interruptions. This is precisely what safe interruptibility aims at avoiding, and we believe this could lead to many problems in a fully adversarial setting in which no model can be learned for interruptions.

## C.5   Pruning mechanism

The pruning mechanism for IDLs is strong – but, to our knowledge, it is necessary to ensure safe interruptibility. However, in practice, states and rewards are not based on the actions of all the agents, but only on the actions of a subset of them. It would then be possible for agents to prune only the observations for which one of their neighbors (the ones their rewards depend on) is interrupted. Therefore, less actions would be pruned and the setting would become practical again. However, since different (but potentially neighboring) agents would learn on different observations, this may make convergence to equilibrium (which is not guaranteed theoretically) even more unlikely.