[Reviews · NeurIPS 2017]

Reviewer 1



This paper presents an extension of the safe interruptibility (SInt) framework to the multi-agent case. The authors argue that the original definition of safe interruptibility is difficult to use in this case and give a more constrained/informed one called 'dynamic safe interruptibility' (DSInt) based on whether the update rule depends on the interruption probability. The joint action case is considered first and it is shown that DSInt can be achieved. The case of independent learners is then considered, with a first result showing that independent Q-learners do not satisfy the conditions of the definition of DSInt. The authors finally propose a model where the agents are aware of each others interruptions, and interrupted observations are pruned from the sequence, and claim that this model verify the definition of DSInt. The paper is mostly well-written, well motivated, offers novel ideas and appears mostly technically correct. The level of formalism is good, emphasizing on the ideas rather than rendering a boring sequence of symbols, but this also hurts somewhat the proof reading of the appendix, so I can't be 100% confident about the results. * Main comments: - What is the relation between the authors' definition of DSInt and the original version? In particular, do we have DSInt implies SInt, or the converse, or neither? - In the proof of Lemma 1, p. 12, third equality, a condition on 'a' is missing in the second P term. As far as I understand, this condition cannot be removed, preventing this term from being pulled out of the sum. * Minor comments: - The motivating example is pretty good. On a side note however, it seems that in this scenario the knowledge of others' interruptions may still be inadequate, as in the general case the different vehicles may not even be of the same brand and thus may not exchange information (unless some global norm is defined?). This is not so much of a criticism, as one needs to start somewhere, but a discussion about that could go in the conclusion. - Definition 2: This should define "dynamic safe interruptibility", not just "safe interruptibility" which is already defined in [16]. - The notation used for open intervals is at times standard (in the appendix) and at other times non-standard (as in the main text). This should be made homogeneous (preferably using the standard notation). - Def 5: "multi-agent systems" -> "multi-agent system" - Proof of Thm 2: On the first equality in the display, we lost the index (i). - The second line after the display should probably read \hat{Q^m_t} instead of Q^{(m)}_t. - Missing period at the end of the proof. - Proof of Thm 3: "two a and b" -> "two agents a and b" (or learner, player?) - \gamma is not defined AFAICT - Proof of Thm 4, "same than" -> "same as" - Does it mean that independent learners are DSInt but not SInt? - Penultimate sentence of the conclusion, "amount exploration" -> +of - p.12 after Lemma 2 "each agents do not" -> "each agent does" or "no agent learns" - last line of the display in the proof of Lemma 1: Use the \left[ and \right] commands (assuming LaTeX is being used) to format properly. - proof of Thm 4: q is undefined.

Reviewer 2



Summary: The paper proposes a definition of safe interruptibility (SI) in the multi-agent setting. SI means that a given RL algorithm can still learn the optimal policy while experiencing (adversarial) interruptions: instances in which an external policy pi_int is used instead of the policy that is being learned. One interpretation is that agents need to explore all state-actions and converge to the same optimal policies / Q-value fixed points, even when interruptions are present - hence SI comes down to ensuring that learning / exploration is defined in a way that is "independent enough" from historically executed actions / interruptions. In the (competitive) multi-agent setting, convergence to optimal Q-values is not guaranteed, hence the paper defines SI dynamically - i.e. learning with SI should preserve dynamics, the probability of seeing a particular Q-value during training should be unchanged. The paper then studies DSI for Q-learning with e-greedy style learning: using joint learners (Q-value function of all states and all actions) and independent learners (Q-value depends on all states and only agent's action). In particular, it shows that certain schemes to set epsilon during training are robust to interruptions. The paper then concludes that JAL is DSI with a well-chosen e-scheme, and IDL similarly is DSI, given an additional pruning mechanism that removes experience where an agent was interrupted. As such, the paper proposes an interesting theoretical extension of RL and raises new questions to pursue. However, the work only considers a very limited setting and the practical implications of the current work are not illustrated. Additional concerns / comments: - The paper considers tabular learning only (no function approximations) and so the analysis seemingly breaks down e.g. for infinite # of states. Given function approximators are essential for most applications, what is the biggest obstacle to applying the DSI analysis to that case? - The analysis does not apply to on-policy methods, as the paper notes. This seems like a major omission, although it might be out of the scope of this paper. How could we verify SI in the on-policy case (e.g. use a strong form of entropy regularization)? - The paper notes that the defined e-scheme is robust to the most adversarial interruption scheme possible. It would be nice to have some (synthetic) experiments to showcase a practical implementation to see how much better it can do in practice. - How precisely could learning a model of the interruption process (i.e. learning theta) help? - The pruning mechanism for IDL seems overly strong - how does the analysis change if agents learn to communicate / share their model of the interruption process?

Reviewer 3



The paper proposes a method to extend previously-published framework of safe interruptibility to multi-agent systems. Formal requirements for safe dynamic interruptible systems are provided and theoretical guarantees are given when systems of jointly learning or independently learning agents are safely interruptible. The paper is well-motivated and exposition is clear. The concrete example of self-driving cars is valuable and clearly illustrates the need for the research in this area. The proposed theoretical foundations seem sound, although I am not an expert in this area. The provided proofs are useful as their techniques could be of value to other researchers who would like to continue work in this area. Ideally, I wish the authors would provide the readers with more tools to extend their theoretical work, such as general guidelines on how to prove safe interruptibility for other learning algorithms (approaches that rely on anticipating strategies of other agents, for example, which are mentioned not to be safely interruptible) I think this paper tackles an important problem. The preliminary paper on safe interruptibility opened many avenues of research to explore. The authors of this paper have picked an important extension of safe interruptibility in multi-agent systems and have done a solid theoretical investigation that I believe will be of value to the machine learning community and I would like to see it published at NIPS.